# A Region-Monitoring-Type Slitless Imaging Spectrometer

**DOI:** 10.3390/s24134242

**Published:** 2024-06-29

**Authors:** Rui Ouyang, Duo Wang, Longxu Jin, Tianjiao Fu, Zhenzhang Zhao, Xingxiang Zhang

**Affiliations:** 1Changchun Institute of Optics, Fine Mechanics and Physics, Chinese Academy of Sciences, Changchun 130033, China; ouyangrui16@mails.ucas.edu.cn (R.O.); jinlx@ciomp.ac.cn (L.J.); futianjiao@ciomp.ac.cn (T.F.); jluzzz@163.com (Z.Z.); 2University of Chinese Academy of Sciences, Beijing 100049, China; 3School of Electronic Information and Electrical Engineering, Shanghai Jiao Tong University, Shanghai 200240, China; wangduoup@sjtu.edu.cn

**Keywords:** spectrometer, imaging, region monitoring

## Abstract

In modern scientific practice, it is necessary to consistently observe predetermined zones, with the expectation of detecting and identifying emerging targets or events inside such areas. This research presents an innovative imaging spectrometer system for the continuous monitoring of specific areas. This study begins by providing detailed information on the features and optical structure of the constructed instrument. This is then followed by simulations using optical design tools. The device has an F-number of 5, a focal length of 100 mm, a field of view of 3 × 7, and a wavelength range spanning from 400 nm to 600 nm. The optical path diagram demonstrates that the system’s dispersion and imaging pictures can be distinguished, hence fulfilling the system’s specifications. Furthermore, the utilization of a Modulation Transfer Function (MTF) graph has substantiated that the image quality indeed satisfies the specified criteria. To evaluate the instrument’s performance in the spectrum observation of fixed regions, a region-monitoring-type slitless imaging spectrometer was built. The equipment has the capability to identify a specific region and rapidly capture the spectra of objects or events that are present inside that region. The spectral data were collected effectively by the implementation of image processing techniques on the captured photos. The correlation coefficient between these data and the reference data was 0.9226, showing that the device successfully measured the target’s spectrum. Therefore, the instrument that was created successfully demonstrated its ability to capture images of the observed areas and collect spectral data from the targets located within those regions.

## 1. Introduction

In the modern world, it is frequently necessary to consistently observe a specific region and promptly and precisely recognize objects or events. In the field of astronomy, continuous observations of a certain area make it possible to identify quick variations in specific measurements within a brief timeframe. This, in turn, allows for making deductions about the underlying structures of that region. In military circumstances, the purpose of surveilling a particular region is to swiftly identify any hidden enemy movements.

Researchers have proposed numerous optical identification schemes in pursuit of more precise target identification by scrutinizing various parameters within a given area. Among them, spectrometers stand out as critical instruments in analytical applications and have demonstrated extensive usage across diverse fields [1], such as chemistry [2], biology [3], food science [4], and agriculture [5,6]. This instrument provides a continuous and detailed spectral dataset, allowing for a thorough investigation of the optical reflectance spectrum. This capability enables the accurate identification and characterization of different materials. Spectral analysis enables the examination of the frequencies and intensities of emitted and absorbed light radiation by substances, allowing for qualitative, quantitative, and structural evaluations. This technique has a broad range of applications.

Currently, there are two main methods available for continuously observing and collecting spectral data from targets in a particular area. One technique entails applying a layer on telescopes to collect data across multiple distinct wavelength ranges. Simultaneously, the alternative approach uses conventional dispersion imaging spectroscopy to scan and collect the spectral data from the designated region. In this context, we discuss these methodologies by employing astronomy as an illustrative case.

The telescope-coating approach is the most prominent solution. The Atmospheric Imaging Assembly (AIA) equipment, part of the Solar Dynamics Observatory (SDO) project announced in 2011, conducted thorough observations of the sun using certain wavelength bands. These bands include 450 nm, 170 nm, 304 nm, 160 nm, 17.1 nm, 19.3 nm, 21.1 nm, 33.5 nm, 9.4 nm, and 13.1 nm. Manufacturers have chosen to apply telescope coating in order to enhance the process of observing targets. As depicted in Figure 1, the application of various coatings to the mirrors of four telescopes resulted in three telescopes successfully capturing images in six distinct extreme ultraviolet bands. On the other hand, one of the telescopes had limited capability and could only conduct spectral imaging in four specific ultraviolet bands [7].

The utilization of telescope coatings has resulted in substantial progress. For example, in the AIA system, the application of coated mirrors allows for imaging in various ultraviolet (UV) wavelengths, hence improving its ability to see different phenomena. This strategy enables the precise identification and examination of particular events or targets within a specified area of interest. Like the AIA, various additional devices have been created to consistently record images of certain areas using telescopes that have been coated. This coating enables these devices to collect data over multiple distinct wavelength ranges. The benefit of these devices lies in their capacity to quickly determine the spatial coordinates of desired targets within a given area and collect spectral data across several wavelengths. Nevertheless, they are unable to acquire uninterrupted spectra of the desired targets and frequently experience reduced spectral resolution. As an example, the AIA system’s coating achieves a spectral resolution of only 0.5 nm.

As an example of the dispersion imaging spectroscopy technique, we looked at the extreme ultraviolet imaging spectrometer (EIS) that is present on board the Hinode Sunrise satellite. The Hinode satellite, launched in 2007, was jointly developed by Japan, the United Kingdom, and the United States. It is equipped with an EIS (Extreme Ultraviolet Imaging Spectrometer) to study the emission lines of the solar corona and transition area. The EIS encompasses two consecutive spectral regions, specifically, 17–21 nm and 25–29 nm. The instrument utilizes a highly sophisticated configuration in which the incident light enters the system by an opening, travels through filtering, is concentrated by a main mirror onto a narrow opening, and is subsequently filtered again before entering a curved diffraction grating. Subsequently, the light that has undergone diffraction is concentrated onto two detectors. The principal mirror and grating are split into two halves, with each piece covered in a specially designed Mo/Si film. This coating guarantees that the mirror, grating, and detectors are properly aligned with each other [8]. The instrument’s aperture is partitioned into two bands, each of which produces images for its own band. The two spectral bands, specifically, the short wavelength band ranging from 166 to 212 Å and the long wavelength band ranging from 245 to 291 Å, offer precise measurements of electron density, temperature, emission measures, and elemental abundance [9]. The optical path is depicted in Figure 2 (according to the journal article as [10]).

Devices resembling the EIS have the capability to acquire spectral data with an exceptionally high spectral resolution. Nevertheless, these instruments are constrained by the existence of a narrow slit, which greatly limits the range of vision and hinders complete observations of the entire frame. This can result in failing to achieve objectives.

Both the coated telescope approach and the dispersion-type imaging spectrometer approach have constraints in terms of observing a defined region, identifying targets of interest, and acquiring target spectra. The coated telescope method is unable to acquire uninterrupted spectra, but the dispersion-type imaging spectrometer is incapable of concurrently observing the entire frame.

This paper presents an apparatus that has been developed to offer spectrum imaging capabilities for a specific region, with the aim of improving continuous monitoring, target detection, and identification. The equipment is specifically designed for the spectral imaging of specific regions. It allows for the quick identification and mapping of targets or events within the area of interest, while simultaneously capturing their unique spectral characteristics.

## 2. Principle of the Region-Monitoring Imaging Spectrometer

### 2.1. Application Areas of the Instrument

This study presents an instrument specifically built for predictive applications in situations that require constant monitoring of a specified area, such as waiting for targets to materialize in that area. The main emphasis is on small targets or phenomena, which typically exhibit specific traits such as sporadic occurrences, brief durations, and challenges in extracting their features. By being able to comprehensively observe the full field of vision and quickly capture the spectra of targets that occur in the area, it would assist in extracting their characteristics, thus improving the target recognition abilities.

Currently, the widely used spectral technologies, including slit-type imaging spectrometers, Fourier transform imaging spectrometers, and filter-type imaging spectrometers, all have their limitations in this field. Slit-type imaging spectrometers require scanning the entire field of view to detect the spectra of the entire area, but this scanning process is prone to missing phenomena. Fourier transform imaging spectrometers require data from one phase cycle of the target. Therefore, it is difficult to obtain target spectra promptly when the target appears for only a short time. Filter-type spectral technologies can observe targets promptly but can only obtain spectra data from a few discrete channels.

Hence, this paper investigates a spectrometer that is capable of simultaneously observing the entire area while rapidly acquiring the spectral data from the target of the area within a short time.

### 2.2. Overview of the System

The device discussed in this article must have the ability to consistently monitor a designated region and collect the spectral data of any targets that occur inside that region. In order to achieve this objective, the device described in this article serves two purposes: the first is the conventional imaging function, while the second is to acquire a dispersion graph of the targeted region. The dispersion graph is defined as an image after the image has been dispersed by a dispersion element, which will be explained in more detail later in the text.

The traditional imaging graph allows for the observation of the positions and shapes of specific targets or phenomena in the target region. Meanwhile, the dispersion graph enables the observation of spectrum changes that occur as a result of the appearance of these objects. By utilizing this technique, it is possible to acquire spectral data about objects present in the vicinity while maintaining a constant observation of the area. The procedure is illustrated in Figure 3 and Figure 4.

### 2.3. Principle of the System

The developed instrument produces two images of the observed area: conventional imaging graph and dispersion graph. The dispersion graph depicts the dispersion of the conventional graph along a certain direction. For simplicity, let us assume a series of object points $y_{i}$ each characterized by wavelengths $\lambda_{j}$, where i ranges from 1 to s, and j ranges from 1 to n. Figure 5 depicts the schematic of the conventional imaging graph, while Figure 6 presents the schematic of the dispersion graph, with each grid representing a pixel.

When a phenomenon occurs at point $y_{i}$, the luminance of that point must have changed. At the same time, the brightness of each of the $\lambda_{1}$–$\lambda_{n}$ bands at that point has thus also changed. The regions of change in the two system images are illustrated by the red boxes in Figure 7 and Figure 8. Assuming the brightness change is $\delta_{y}$, and that the spectral change in each band is $\lambda_{1y}$–$\lambda_{ny}$ (as depicted in Figure 7), the difference between the pre-phenomenon and post-phenomenon imaging graph then represents the brightness change. Similarly, the difference between the dispersion graph before and after the phenomenon indicates the variations in each spectral band, which are denoted as δλ.

By subtracting the images captured before and after the occurrence of the phenomenon, two new images can be obtained as displayed in Figure 9 and Figure 10.

We hypothesized that a flare would occur within a designated region. Based on empirical knowledge, flares exhibit significant spectral variations across certain bands. Thus, from Figure 9, we can pinpoint the spatial location of the flare, as well as all the variations concentrated within a single pixel. Additionally, Figure 10 allowed us to quantify the spectral changes at that point, as those variations were distributed across a series of pixels [11].

### 2.4. The Spectral Resolution of Instruments

The spectral resolution of a spectroscopic system is the core capability of the instrument that is of the utmost concern. In the realm of the area-monitoring imaging spectroscopy systems studied in this paper, spectral resolution is determined not only by the system itself, but also by the width of the target’s dispersion in the image. This is because traditional dispersive imaging spectrometers have a fixed slit, whereas such a slit does not exist in area-monitoring imaging spectrometers. Therefore, the width of the object (or phenomenon) in the image plays a role similar to the slit in traditional spectrometers.

Assuming that the grating constant of the instrument is *d*, the grating order is m, the incidence angle is θ, the exit angle is $\theta_{r}$, the pixel width of the instrument is *p*, the minimum band width that the instrument can distinguish is δλ, and the minimum diffraction angle that the instrument can distinguish is $\delta \theta_{r}$. According to the grating equation, Equation (Equation 1) can be obtained.
(1)$$\lambda =\frac{d(sin\left(\theta \right)-sin\left(\theta_{r}\right))}{m};$$

Equation (Equation 2) can be obtained by simultaneously differentiating left and right.
(2)$$\delta \lambda =\frac{d\left(cos\left(\theta_{r}\right)\right)\times \left(\delta \theta_{r}\right)}{m};$$

Therefore, the spectral resolution of the instrument *L* is shown in Equation (Equation 3).
(3)$$L=\frac{\lambda }{\delta \lambda }=\frac{\left(sin\left(\theta \right)-sin\left(\theta_{r}\right)\right)}{\left(\delta \theta_{r}\right)\times \left(cos\left(\theta_{r}\right)\right)};$$

Assuming that the target covers a pixel in the dispersion direction, with each pixel width *p*, the minimum diffraction angle $\delta \theta_{r}$ that the instrument can distinguish is shown in Equation (Equation 4).
(4)$$\delta \theta_{r}=arctan\left(\frac{ap}{f}\right);$$

Therefore, the spectral resolution of the instrument can be simplified to Equation (Equation 5).
(5)$$L=\frac{\left(sin\left(\theta \right)-sin\left(\theta_{r}\right)\right)}{arctan\left(\frac{\left(a)\times (p\right)}{f}\right)\times \left(cos\left(\theta_{r}\right)\right)};$$

Equation (Equation 5) is the method for calculating the spectral resolution of the instrument. Due to the fact that in reality, (a)×(p) is much smaller than f. Therefore, it can be considered that there exists an approximate relationship $arctan\left(\frac{\left(a)\times (p\right)}{f}\right)\approx \frac{\left(a)\times (p\right)}{f}$. So, *L* can also be written as Equation (Equation 6).
(6)$$L=f\times \frac{\left(sin\left(\theta \right)-sin\left(\theta_{r}\right)\right)}{\left(a\right)\times \left(p\right)\times (cos\left(\theta_{r}\right))};$$

Finally, the spectral resolution can be simplified to Equation (Equation 7).
(7)$$L=\frac{\left(f)\times (\lambda )\times (m\right)}{\left(d\right)\times \left(a\right)\times \left(p\right)\times (cos\left(\theta_{r}\right))};$$

## 3. The Optical Design of the Region-Monitoring Imaging Spectrometer

Given the discrepancies that exist between the developed instrument and traditional imaging spectrometers, we detail here our instrument’s optical design process. As outlined earlier, the developed instrument requires a conventional imaging image and a dispersion image, both at the same wavelength. Thus, a grating-based approach was considered, where the 0th order of the grating produces the conventional imaging image, and the ±1st orders produce the dispersion image. The instrument should be designed to facilitate the placement of a field stop and to maintain good image quality.

The optical design requirements for such an instrument are proposed as follows:The instrument must simultaneously produce both images without overlap.The image quality must meet the specified standards. Assuming a 5 μm CMOS, the point spread function at a 600 nm wavelength must be within 5 μm. Consequently, the system’s F-number must be less than 5.The system requires an intermediate image plane to accommodate a square field stop, thus isolating the dispersion and imaging images without significantly increasing the overall system length.The image quality must meet the specified standards.The overlapping of spectra of different orders should be avoided.

Based on these requirements, we designed the structure illustrated in Figure 11, Figure 12 and Figure 13. The different colors in these images represent different fields of view.

Figure 11 depicts the zeroth-order spectrum image produced by the grating, while Figure 12 illustrates the first-order spectrum image. According to the grating equation, when the grating order is 0, the system exhibits no dispersion, and the incident angle equals the reflection angle, thus resulting in normal imaging. Conversely, the first-order spectrum introduces dispersion, forming a dispersion image. Figure 13 depicts the system’s imaging, where the zeroth- and first-order spectra are simultaneously captured. The lens portion of the system can be custom designed or purchased to suit specific requirements.

Figure 14 and Figure 15 show the MTF of the dispersion graph, and it can be seen that the MTF of the proposed system was higher than 0.6 at all 40 lines; as such, the image quality of the system was considered to meet the requirements.

Table 1 shows the system parameters.

## 4. Experimentation and Data Processing

Due to the low diffraction efficiency of the grating of the instrument, the dispersion graph of the instrument was much less bright than the imaging graph; as such, a light source was used to fill the detection area with light. After setting up the instrument, the instrument was used to image the detection area with the following process: use any small object as a target and put it into the detection area to detect the spectral data. Measure the spectra of the target using a professional spectrometer under the same light conditions. Finally, the correlation between the comparison data measured by the professional spectrometer and the detection data of the instrument was calculated to verify the spectral detection capability of the instrument for the target in the area.

### 4.1. Experimentation

A region-monitoring imaging spectrometer was constructed and tested to validate our instrument’s performance. Figure 16 depicts the physical layout of the optical system. The components presented in Figure 16 include the instrument’s entrance pupil, filter, parabolic mirror, reflectors, grating, imaging lens, and CMOS camera.

The experimental observations as depicted in Figure 17 involved the continuous monitoring of a specific region as displayed in Figure 17a. Subsequently, the appearance of a target is illustrated in Figure 17b. Following the emergence of a target, variations in the spectral region were observed, and the disparity that was captured before and after the target’s appearance is illustrated in Figure 17c.

The X-axis and Y-axis in Figure 17c are the number of pixels, the blue color in the background is the region with almost zero value, and the pattern in the middle part is the difference between the brightness of the imaging map and the dispersion map before and after the target appears.

### 4.2. Instrument Calibration

The instrument was calibrated using a low-pressure mercury lamp, whose physical characteristics and spectrum are depicted in Figure 18. Calibration served to establish the correspondence between points in the imaging map and their counterparts in the dispersion diagram, thereby enabling the measurement of the instrument’s response across different wavelengths. Additionally, it was necessary to ascertain the regions in the dispersion diagram corresponding to the points within the field of view. The low-pressure mercury lamp exhibited four distinct peaks at 404 nm, 435 nm, 546 nm, and 570 nm, where each peak’s intensity aided in the inversion of the actual images.

Figure 19 illustrates the luminance curve that was obtained from multi-point measurements. The left side represents the spatial positions of the low-pressure mercury lamp, while the right side depicts the response within the dispersion diagram.

### 4.3. Data Processing

The spectral data of the target were obtained under identical brightness conditions using a specialized spectrometer as shown in Figure 20.

The spectral data of the target were acquired under the same level of brightness using a specialized spectrometer as depicted in Figure 20.

The moving average method is a commonly used technique for smoothing time series data. This technique reduces random fluctuations and reveals the underlying trends or cyclical changes in the data. It works by averaging multiple points in the input signal to produce each point in the output signal.

Assuming the original time series is ($x_{1}$, $x_{2}$, …, $x_{n}$), and the window size for the moving average is (*m*) then the moving average $f_{t}$ at time point t can be calculated using the following Equation (Equation 8):(8)$$f_{t}=\frac{1}{m}\sum_{i=t-m+1}^{t}x_{i}.$$

Wavelet denoising is a classical signal processing technique. The basic idea of this technique is to realize the removal of noise components and to leave the key information through decomposition and selective reconstruction.

The original Signal *x*(*t*) was decomposed into an approximate Signal A and a detailed Signal D using the wavelet transform; as such, we obtained *x*(*t*) = *A* + *D*.

A threshold function T was applied to the detail Signal (*D*) to remove or minimize the coefficients that were considered noise: $D_{denoising}$ = *T*(*D*).

When reconstructing the denoised signal ($x_{denoising}\left(t\right)$) using the denoised approximation coefficients (A) and the thresholded detail coefficients ($D_{denoising}$), we used the following equation: [$x_{denoising}\left(t\right)=A+D_{denoising}$].

MATLAB software 2017 version was used to denoise the noisy spectral data in Figure 21. MATLAB’s function movmean was used to perform the moving average of the curve, where the window size was 8. MATLAB’s function wdenoise was used to conduct the wavelet denoising of the curve, where the denoising level was 5 and the base of the wavelet transform was blockj. This combination of programs represents a tested and more effective manner, through which to eliminate the curve’s random noise. The curve of Figure 22 was then finally obtained.

The Pearson correlation coefficient is a statistical metric employed to evaluate the degree of linear association between two variables. On a scale from −1 to 1, it measures the extent of the linear relationship by calculating the covariance of the two variables. A coefficient of 1 signifies a flawless positive correlation, −1 signifies a flawless negative correlation, and 0 signifies the absence of a linear relationship between the variables. Typically, a score exceeding 0.8 signifies a robust correlation between the two datasets.

Suppose there are two arrays X and Y; then, ρ(X,Y) is the Pearson’s correlation coefficient between the two arrays, which is calculated with the following Equation (Equation 9):(9)$$\rho \left(X,Y\right)=\frac{E(XY)-E(X)E(Y)}{\sqrt{E\left(X^{2}\right)-E{\left(|X|\right)}^{2}}\times \sqrt{E\left(Y^{2}\right)-E{\left(|Y|\right)}^{2}}}.$$

The calculated Pearson correlation coefficient between the comparison item and the denoised data was 0.9226, thus indicating that the instrument successfully measured the target’s spectral data.

## 5. Results

We have constructed a region-monitoring imaging spectrometer, and the specific characteristics of the device can be found in Table 1. The equipment consistently captures photos of a specific area. The presence of a target in the area causes a modification in both the conventional image and dispersion graph. Consequently, we obtained the necessary spectral data by utilizing two dispersion graphs, one taken before and one taken after the target’s appearance. The spectral data were heavily contaminated with noise. In addition, we employed a denoising method that integrates wavelet modification and moving average to acquire data that were effectively filtered of noise. The Pearson’s correlation coefficient between the denoised data and the reference data was 0.9226, showing a robust statistical correlation between the two datasets. Thus, we can deduce that the CMOS sensor efficiently records the spectrum information of the target.

## 6. Discussion

This study investigates a region-monitoring imaging spectrometer. We will now discuss the results in terms of three aspects.

### 6.1. Discussion of Measurement Results

Firstly, regarding the measurement of the target spectral data, although the calculated Pearson coefficient showed a strong correlation with the denoised and the reference data, there were visual discrepancies. We attributed these differences to several factors:Vignetting, which affects the measurement outcome.Random noise from the CMOS camera in both measurements, which affects measurement accuracy.The instrument’s spectral resolution is related to the target’s width in the dispersion dimension. Due to the broader width of the target in the dispersion dimension, the spectral resolution is lower than that of the reference, thus resulting in less distinct peaks in the denoised data.

### 6.2. Discussion of Instrument Advantages

Subsequently, we will now discuss the advantages of the developed instrument over traditional slit-based imaging spectrometers:Higher energy utilization efficiency in the dispersion element. A substantial proportion of energy, around 60%, is concentrated in the zero-order spectrum for gratings. Conventional imaging spectrometers that use slits do not make use of the zero-order spectrum for imaging, whereas the region-monitoring imaging spectrometer completely exploits it.Higher energy utilization efficiency over the full field of view. Traditional slit-based imaging spectrometers can only utilize energy within the slit area due to the presence of the slit. In contrast, the region-monitoring imaging spectrometer can utilize energy across the entire field of view.Lower manufacturing and tuning complexity. Conventional imaging spectrometers typically consist of a front telescope and a rear spectrometer, which utilize a slit-based structure. For example, in astronomy, a regularly used telescope is the off-axis, three-mirror telescope, which necessitates three large-aperture, non-spherical mirrors. The telescope and spectrometer each individually correct aberrations, resulting in an intricate configuration. On the other hand, the region-monitoring imaging spectrometer has the ability to capture images on its own. The telescope and spectrometer’s collimation structure is made up of two symmetrical sections of a parabolic mirror, which makes the structure less complex. The production costs and tuning complexity can be greatly reduced by producing a small-aperture transmission imaging system.

### 6.3. Discussion of Instrument Disadvantages

We also aimed to discuss the disadvantages of the proposed instrument compared to traditional slit-based imaging spectrometers:The instrument’s spectral resolution depends on the target width in the dispersion dimension; thus, there is no stable spectral resolution. The instrument’s spectral resolution may be relatively low when faced with larger targets.There are mutual constraints between the instrument’s field of view and wavelength range.The instrument cannot obtain spectral data for every point in the image but rather captures spectral data for points where changes occur within the monitoring range.

## 7. Conclusions

This study examined a slitless imaging spectrometer of the region-monitoring kind. The fundamental principles of the region-monitoring-type slitless imaging spectrometer were explained. The optical design of the system was carefully examined in order to ultimately create a slitless imaging spectrometer for region monitoring. The device functions within the wavelength range of 400–600 nm, has a focal length of 100 mm, has a transmission grating with a density of 150 lines per millimeter, and provides a field of view measuring 3° × 7°. Additionally, a region-monitoring-type slitless imaging spectrometer was constructed to confirm the instrument’s performance. This apparatus allows for the continuous observation of a certain area, detecting the spectral data of the targets present inside that area. The Pearson correlation coefficient between the denoised data and the comparison items was 0.9226, surpassing the necessary value of 0.8, indicating a successful detection of the target spectra. The gadget showcased its capacity to consistently observe a certain region and swiftly capture the uninterrupted spectra of any targets that emerge within the region.

## Figures and Tables

**Figure 1 sensors-24-04242-f001:**
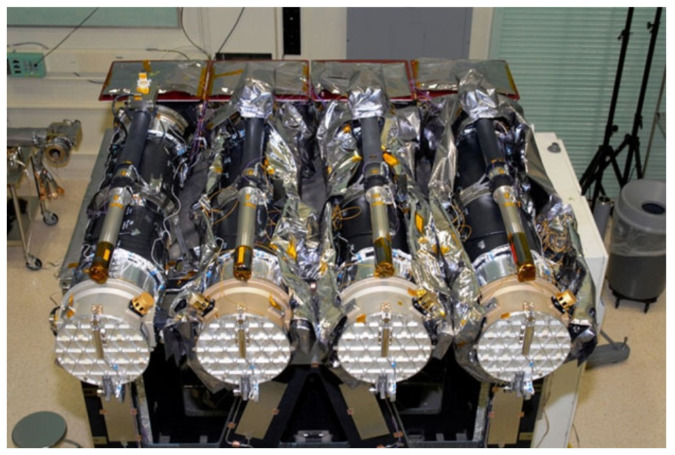
The SDO project setup. All four telescopes adopted mirror coating.

**Figure 2 sensors-24-04242-f002:**
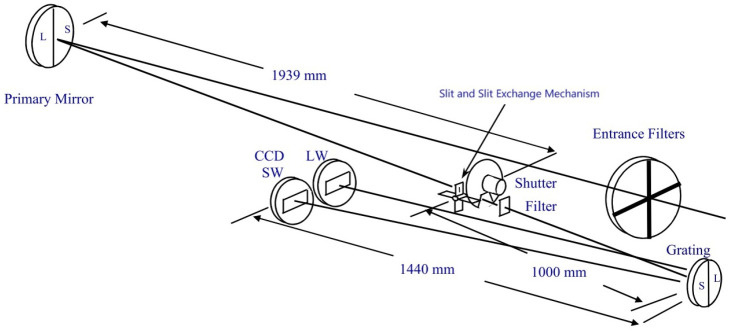
Optical schematic diagram of the extreme ultraviolet imaging spectrometer (EIS).

**Figure 3 sensors-24-04242-f003:**
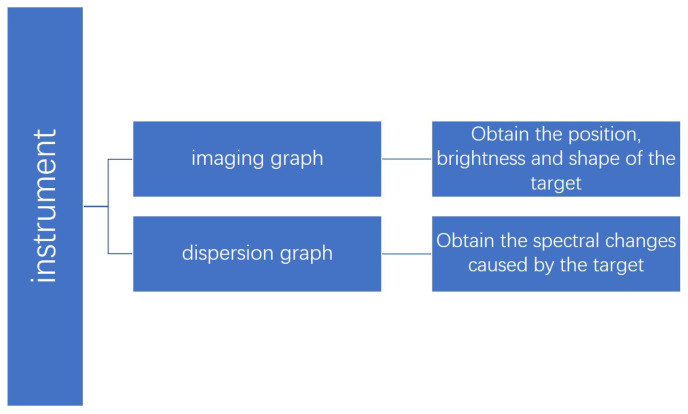
The process of the system.

**Figure 4 sensors-24-04242-f004:**
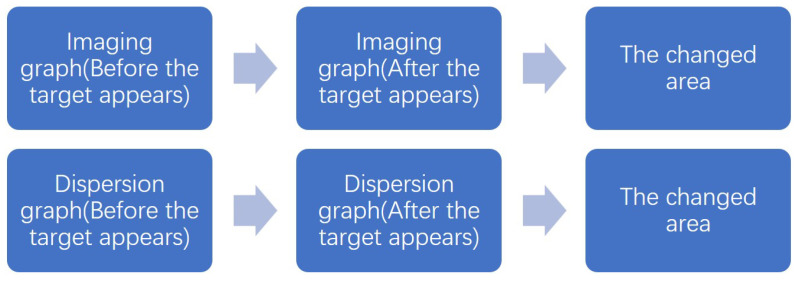
The process of the system.

**Figure 5 sensors-24-04242-f005:**
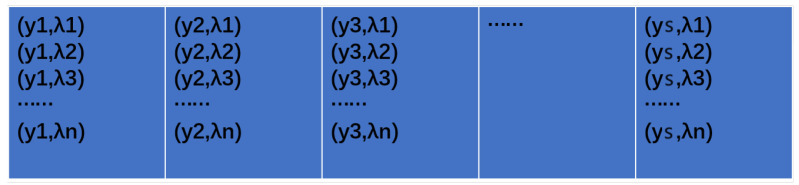
Schematic diagram of the imaging graph.

**Figure 6 sensors-24-04242-f006:**
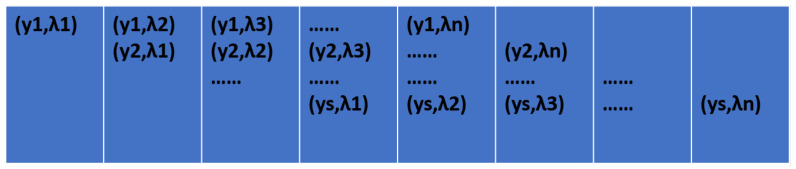
Schematic diagram of the dispersion graph. (This figure was obtained from Figure 5 by dispersion through the dispersion element).

**Figure 7 sensors-24-04242-f007:**
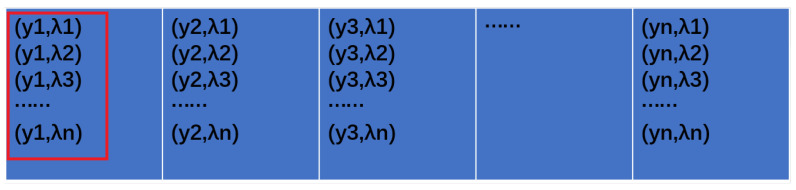
The red boxed area changed.

**Figure 8 sensors-24-04242-f008:**
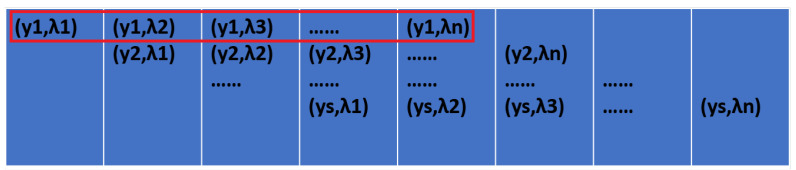
The part of the dispersion graph that changed with Figure 7.

**Figure 9 sensors-24-04242-f009:**
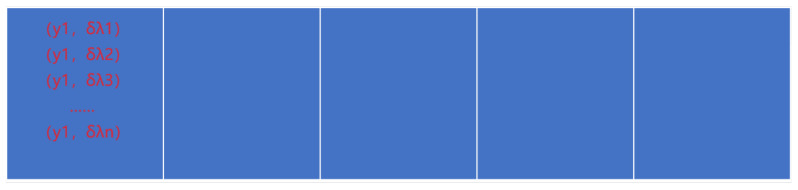
Changes in the imaging graph.

**Figure 10 sensors-24-04242-f010:**
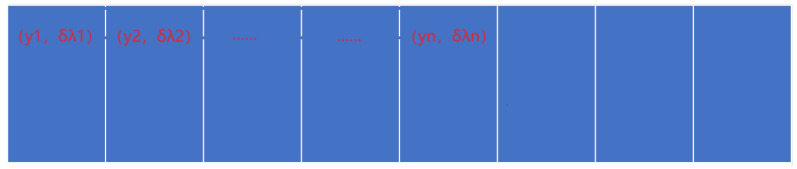
Changes in the dispersion graph.

**Figure 11 sensors-24-04242-f011:**
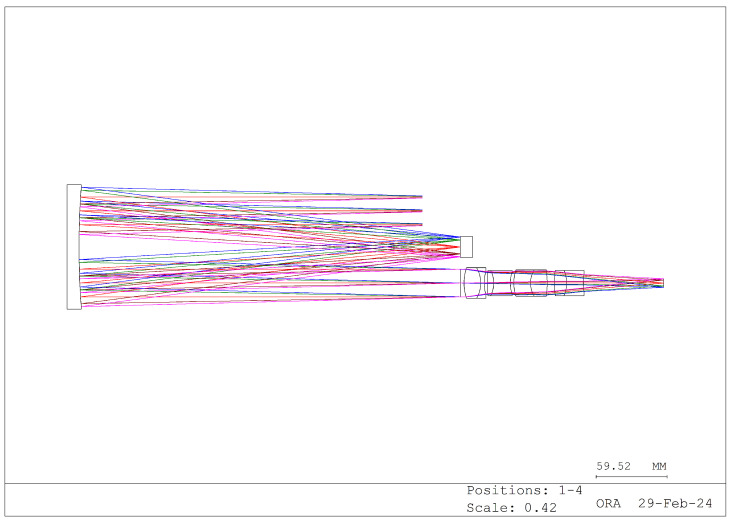
Zeroth-order spectral imaging optical path (imaging graph).

**Figure 12 sensors-24-04242-f012:**
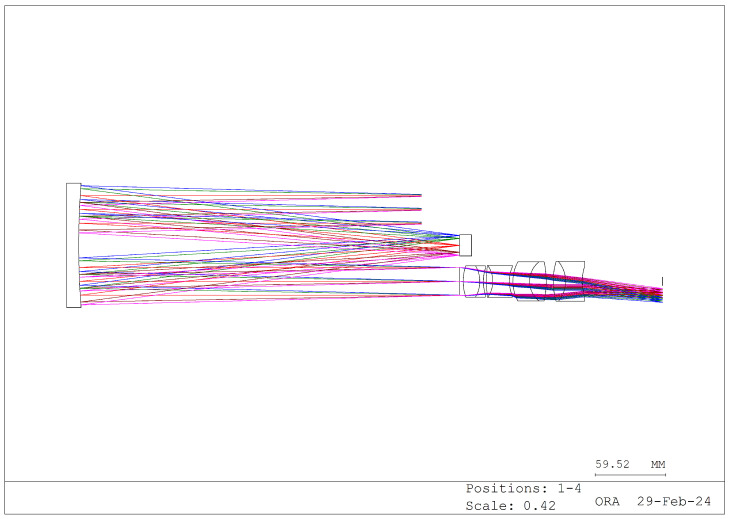
Minus-one-order spectral dispersion optical path (dispersion graph).

**Figure 13 sensors-24-04242-f013:**
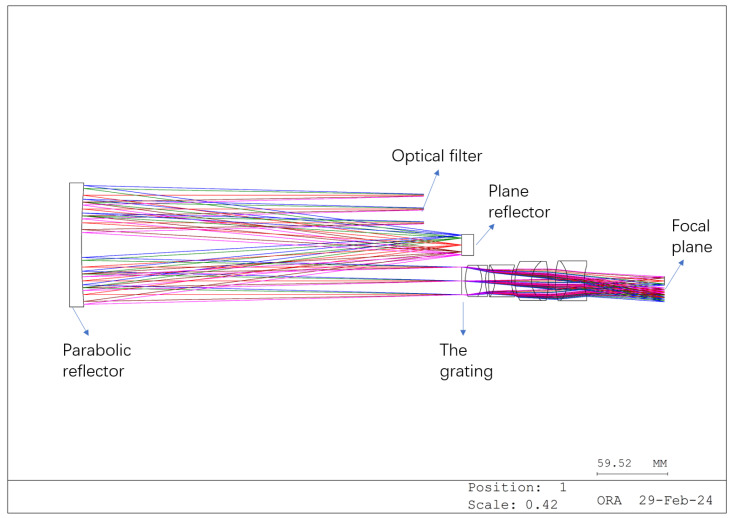
Optical system diagram (different colored lines represent different fields of view).

**Figure 14 sensors-24-04242-f014:**
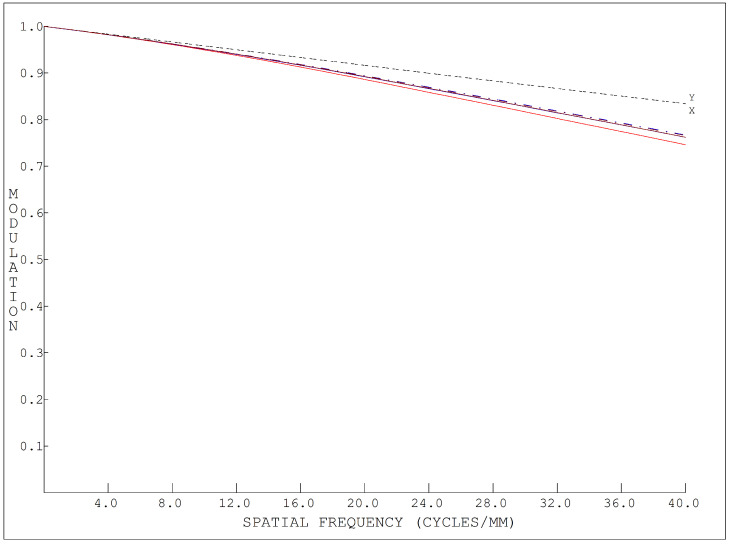
The MTF (Modulation Transfer Function) of the conventional images.

**Figure 15 sensors-24-04242-f015:**
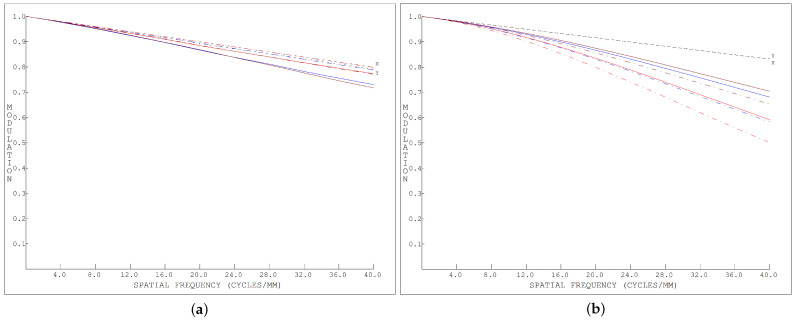
(**a**) 400 nm MTF of the dispersion graph. (**b**) 600 nm MTF of the dispersion graph.

**Figure 16 sensors-24-04242-f016:**
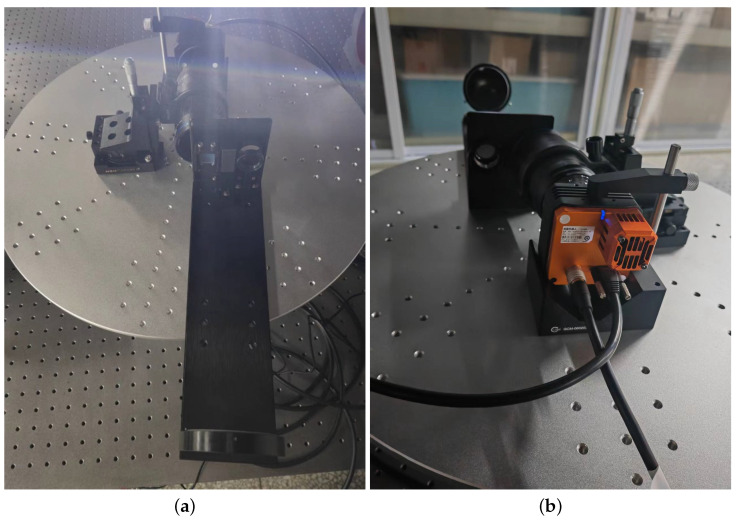
(**a**) Optical system. (**b**) CMOS camera.

**Figure 17 sensors-24-04242-f017:**
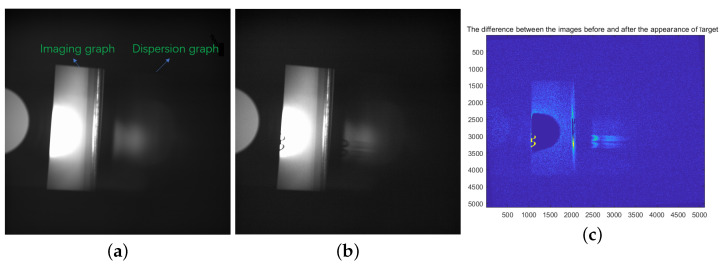
(**a**) Observation area. (**b**) After the target appearance. (**c**) Difference between the two pictures.

**Figure 18 sensors-24-04242-f018:**
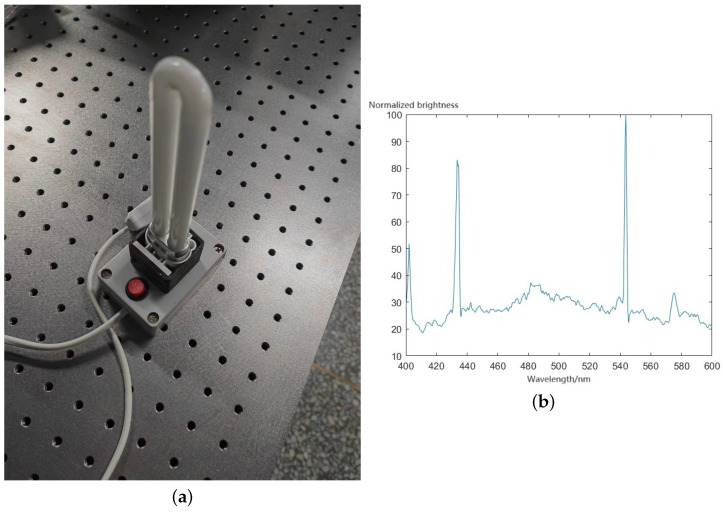
(**a**) Low-pressure mercury lamp. (**b**) The spectrum of the low-pressure mercury lamps.

**Figure 19 sensors-24-04242-f019:**
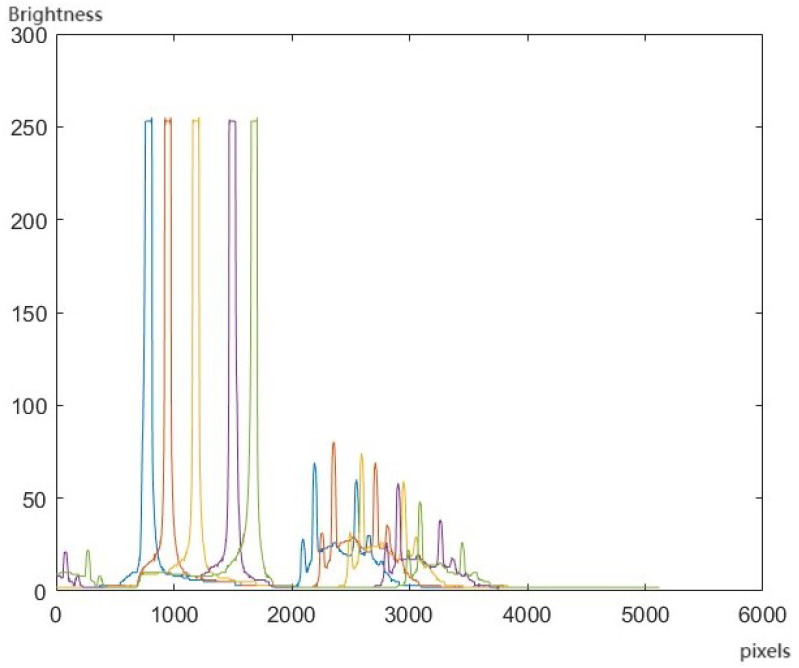
Brightness curve of multiple points.

**Figure 20 sensors-24-04242-f020:**
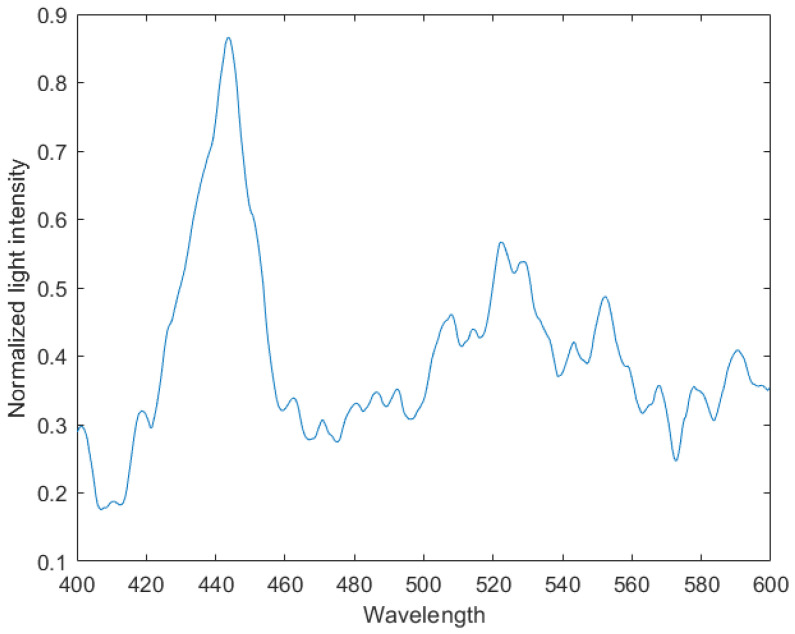
The spectrum of the target (comparison items).

**Figure 21 sensors-24-04242-f021:**
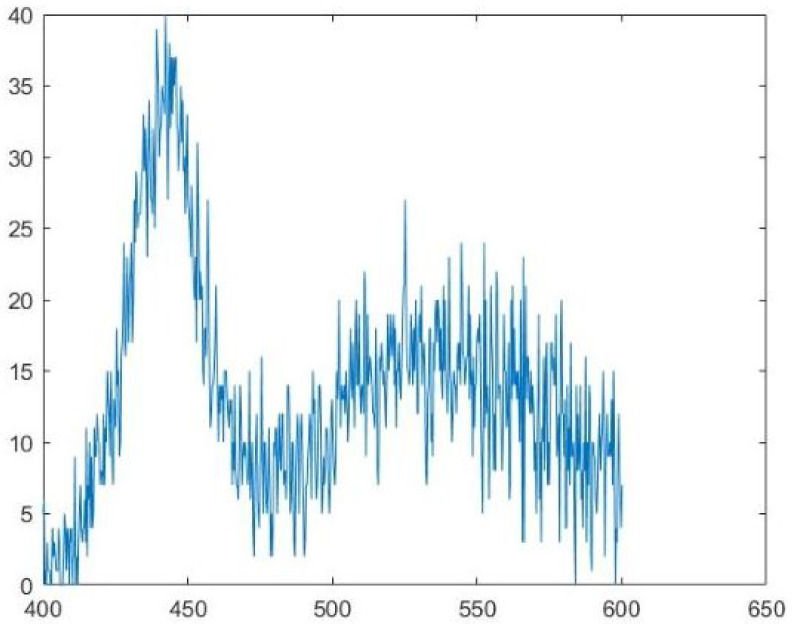
Extracting the raw spectral data from Figure 20.

**Figure 22 sensors-24-04242-f022:**
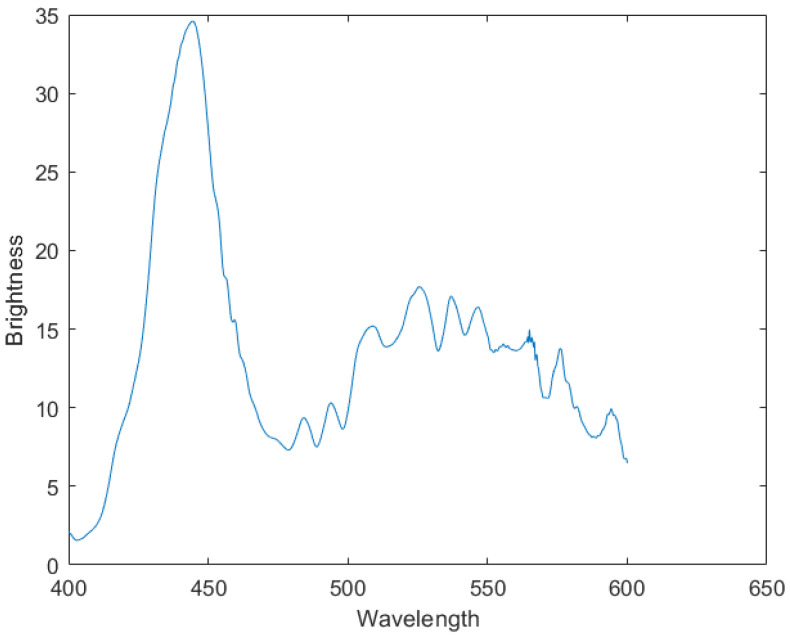
Plot of the denoised spectral data.

**Table 1 sensors-24-04242-t001:** System parameters of the region-monitoring imaging spectrometer.

Parameters	Number
Focal length	100 mm
Bandwidth	400–600 nm
F number	5
Grating type	Transmission
Number of grating line pairs	150 L/mm
Field of view	3° × 7°

## Data Availability

Dataset available on request from the authors.

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
