# Peer review of "A Region-Monitoring-Type Slitless Imaging Spectrometer"

_sensors, 2024, doi:10.3390/s24134242_

Round 1

Reviewer 1 Report

Comments and Suggestions for Authors

In this manuscript, the authors introduce a high spectral resolution imaging spectrometer designed to continuously monitor localized regions. The instrument’s characteristics, optical structure, and optical design software simulations are elaborated upon. Additionally, the system successfully separates dispersion and imaging images. Furthermore, a region-monitoring imaging spectrometer was constructed, successfully obtaining spectral data from fixed areas within the detection range. The high correlation coefficient between acquired data and reference data validates the instrument’s capability.

This manuscript may contain some useful results, however, the experiment procedures and results are poorly organized and presented. The context is not smooth, moderate editing of English language is also required. This manuscript can be reconsidered to publish if the author can address the following several questions:

Major comments:

(1)     In section 1, the concept of the dispersion graph is pretty confusing. Can you have more explanation on it? Is the dispersion graph an image after diffracting from a grating so that one could spatially separate the images which are illuminated by light sources with different wavelengths? How does it correspond to the optical system you designed in section 2.

(2)     It would be much better to have more detailed captions in all figures from 3 to 19 instead of only describing the terms in the main text.

(3)     Figure 3-6 are confusing. Can I understand in such a way that in each tuple (y_i, \lambda_j), each y_i represents a coordinate in the image, while \lambda_j is the wavelength of the light emitted or reflected from point y_i?

(4)     Figure 5 doesn’t contain much extra information, I think figure 3 and 5 can be combined. Same as figure 4 and 6.

(5)     In line 100-101, the authors said, “When a phenomenon occurs at point yi, it can be inferred that the brightness at this point changes, and simultaneously, the spectral range \lambda_1-\lambda_n at this point alters.” Why does the spectral range have to change?

(6)     More detail should be provided in figure 9 to 11. I think the authors should at least introduce what kind of optics are used in the simulation. And what do the colors mean in the figures, do different colors correspond to the light source with different wavelengths? Where is the image plane?

(7)     In section 3 Experimentation and data processing, almost no experimental procedure is presented and discussed. It is hard to keep track of how the experiment was done. What are the target, light source, etc.? More details should be provided.

(8)     What am I looking at in figure 14 and 15? Are they the images captured by the designed optical system? Are they the convention graph and dispersion graph you described in the previous section? What do imaging area and spectral imaging area mean? Are they the places where the image and spectra image show up or the places where you took the image?

(9)     In figure 16, graph legends should be included, say, what does the color represent? What are the x axis and y axis? Are they just pixels number? Etc.

(10) How do you extract the data from figure 17 to get figure 18? More details should be provided.

(11)  In line 156-157, what exactly the denoising strategy you used to get figure 19? More details should be provided.

(12)  In line 158-160, it would be better if the authors can write down a formula of Pearson correlation and detailly describe what variable the authors used to calculate the Pearson correlation.

Minor comments:

(1)     In my opinion, both the abstract and the conclusion should only include the analysis/experiments the authors did and results they got, all the backgrounds should be removed or moved to introduction. For example, “In the contemporary world of science, continuous observation … acquire spectral data from those areas.” should be removed.

(2)     In line 11-12, it seems confused to have the sentence “and a field of view covering the wavelength range from 400nm to 600nm.” since a field of view should be the angular extent of the observable world that is seen at any given moment.

(3)     In line 37, “[]” should be removed or any reference should be included.

(4)     In table 1, what is focal? Should it be focal length? What is F, should it be F-number? What is FOV, is it field of view? All abbreviations should be explained in the paper.

(5)     In figure 13, should the caption be “CMOS camera/detector”?

Overall, I think most of the analysis, experiments and results in this manuscript are poorly presented and not explained. Additionally, the context is not smooth. The manuscript must be improved.

Thank you!

Comments on the Quality of English Language

Moderate editing of English language required.

Reviewer 2 Report

Comments and Suggestions for Authors

The authors considered the concept of an imaging spectrometer, which has the right to exist and will be useful for solving a number of tasks. However, in general, the text of the article has not been significantly improved.

1. The authors presented a very small review and did not justify their decision on the optical scheme in any way.

2. The idea proposed by the authors will work well for small objects that come into view. The larger the object, the more the spectrum will differ from reality.

3. Simultaneous shooting in 0 and +-1 order is more of a problem for the dynamic range of the camera. With a high-quality image in 0 order, the image will be much less bright in 1 order. And in the article, the authors also have to highlight the spectrum from the background spectrum. This will lead to a strong noise in the spectral distribution, which we see in Fig.18.

4. The authors did not justify in any way why they used such an optical scheme with suboptimal PSF.

5. The authors did not provide PSFs for different wavelengths.

I think the article needs to be radically revised!

Reviewer 3 Report

Comments and Suggestions for Authors

The main suggestions are given as follows:

1, the authors mention the MTF curves and its performance in Abstract, but no any MTF result is given in the following context. thus please check and add some necessary figures and its captions.

2, The article proposes "A region monitoring type slitless imaging spectrometer", which has a severe attenuation of spectral resolution due to the absence of slit, e.g., this method is inappropriate for the detection of solar atmospheric activity that requires high spectral resolution. Therefore, it is necessary to give its specific spectral resolution, and what targets are suitable for spectral feature extraction.

3, the slitless spectrometer is not calibrated for spatial and spectral resolution, which is not good for determining its specific application range.

4, The authors should give a more complete and detailed spectral calibration process. In this paper, the spectral data of the target were obtained under identical brightness conditions using a specialized spectrometer, but whether the original spectrum detected in this paper and the spectrum detected by the specialized spectrometer involve alignment and other issues. In addition, the denoising method of the original spectral data is too simple, and the wavelet transformation and moving averages method is introduced in the paper, which needs to be introduced in this part.

5, It is suggested that Figs. 12 and 13, Figs. 14-16 and Figs. 17-19 can be merged into a single figure, which will be more concise.

6. the authors should polish their English presentations for easy reading.

Comments on the Quality of English Language

the authors should polish their English presentations for easy reading

Round 2

Reviewer 1 Report

Comments and Suggestions for Authors

I'm satisfied with the answers and the new version of the manuscript.

However, figure 16(b) should be written in English.

Thank you!

Reviewer 2 Report

Comments and Suggestions for Authors

In the new version, the article looks much more logical. Indeed, there are a number of tasks in which even a not very accurate reflection spectrum of a point object will allow it to be classified. For example, this is how you can distinguish a UAV from a bird. The authors corrected the article based on my comments. But the article still has a very short overview, clearly insufficient for an article of this level!

Reviewer 3 Report

Comments and Suggestions for Authors

No further questions.

Comments on the Quality of English Language

Their English presentations need to be polished further.
